# Defining the molecular mechanisms of the mitochondrial permeability transition through genetic manipulation of F-ATP synthase

Andrea Carrer[1], Ludovica Tommasin[1], Justina Šileikytė[2], Francesco Ciscato[1], Riccardo Filadi[1,3], Andrea Urbani [1], Michael Forte[2], Andrea Rasola [1], Ildikò Szabò[3,4], Michela Carraro[1✉] & Paolo Bernardi [1,3✉]

F-ATP synthase is a leading candidate as the mitochondrial permeability transition pore (PTP) but the mechanism(s) leading to channel formation remain undefined. Here, to shed light on the structural requirements for PTP formation, we test cells ablated for g, OSCP and b subunits, and $\rho^0$ cells lacking subunits a and A6L. Δg cells (that also lack subunit e) do not show PTP channel opening in intact cells or patch-clamped mitoplasts unless atractylate is added. Δb and ΔOSCP cells display currents insensitive to cyclosporin A but inhibited by bongkrekate, suggesting that the adenine nucleotide translocator (ANT) can contribute to channel formation in the absence of an assembled F-ATP synthase. Mitoplasts from $\rho^0$ mitochondria display PTP currents indistinguishable from their wild-type counterparts. In this work, we show that peripheral stalk subunits are essential to turn the F-ATP synthase into the PTP and that the ANT provides mitochondria with a distinct permeability pathway.

[1] Department of Biomedical Sciences, University of Padova, Padova, Italy. [2] Vollum Institute, Oregon Health and Science University, Portland, OR, USA. [3] Consiglio Nazionale delle Ricerche Neuroscience Institute, Padova, Italy. [4] Department of Biology, University of Padova, Padova, Italy. ✉email: carraro.miche@gmail.com; bernardi@bio.unipd.it

The permeability transition (PT) is a $Ca^{2+}$-dependent permeability increase of the mitochondrial inner membrane to ions and solutes with molecular mass up to about 1500 Da[1–3]. It is today generally accepted that the PT is due to the opening of a channel, the PT pore (PTP), as first proposed by Haworth and Hunter in 1979[1–3]. This channel, also called mitochondrial megachannel (MMC), was later identified in patch-clamp experiments in mitoplasts, which defined its maximal conductance (as high as 1.3–1.5 nS) and a number of distinctive, smaller subconductance states[4,5]. The PTP and the MMC are considered to be the same molecular entity because they respond in the same way to the same set of agonists and antagonists[6,7]. The latter include cyclosporin (Cs) A[8–11], which desensitizes the PTP to opening after binding cyclophilin (CyP) D in the matrix[12].

The molecular nature of the PTP is the matter of debate. The first potential candidate has been the adenine nucleotide translocator[1] (ANT), which was later shown to form $Ca^{2+}$-activated channels with conductance of 0.3–0.6 nS that are activated by CyPD and inhibited by ADP[13,14]. The selective ANT inhibitors atractylate[15] (ATR) and bonkgrekate[16] (BKA) have opposite effects on the PT. ATR, which locks the ANT in the "c" conformation (nucleotide binding site facing the cytosol) shows a PT-stimulating effect; while BKA, which locks the protein in the "m" conformation (nucleotide binding site facing the matrix) instead shows a PT-inhibiting effect, suggesting that pore opening and closure could be related to a specific, $Ca^{2+}$-dependent conformational change of the ANT[1]. Given that mitochondria from $Ant1^{-/-} Ant2^{-/-}$ and $Ant1^{-/-} Ant2^{-/-} Ant4^{-/-}$ mice still undergo a CsA-sensitive PT, albeit at increased matrix $Ca^{2+}$ loads;[17,18] and considering that deletion of the $Ppif$ gene (which encodes CyPD) in the $Ant1^{-/-} Ant2^{-/-} Ant4^{-/-}$ background totally prevents the PT[18], at least another CyPD-sensitive channel must mediate PTP formation[19]. The second major candidate for PTP formation is the F-ATP synthase. This hypothesis was put forward after the demonstration (i) that CyPD binds to, and modulates the F-ATP synthase in a CsA-sensitive manner;[20] and (ii) that partially purified F-ATP synthase generates $Ca^{2+}$-activated channels with the features expected of the corresponding PTPs in bovine[21], human[22], yeast[23], and drosophila[24] mitochondria. Investigations based either on knockdown[25] or on selective ablation of individual subunits of F-ATP synthase[26–28] have generated conflicting results, since both persistence[26–28] and inhibition[25,29] of the PT have been reported to occur. In yeast, absence of the "dimerization" subunits e and g, and of the N-terminal segment of subunit b[30], which closely interacts with subunit g[31], dramatically decreases both size of the PTP and channel conductance of F-ATP synthase[32]. Furthermore, point mutations that do not affect either assembly of the enzyme complex or ATP synthesis did cause specific changes in the channel properties of the PTP[22,32–37]. Finally, highly purified F-ATP synthase preparations displayed the features expected of the PTP in electrophysiological experiments[38,39]. In order to address the many open questions about the role of F-ATP synthase in channel formation, we have studied the features of the PTP by in situ techniques and by patch-clamp recordings of mitoplasts deriving from HeLa cells ablated for subunit g ($\Delta$g), from HAP1 cells individually ablated for subunit b ($\Delta$b) and subunit OSCP ($\Delta$OSCP)[28] and from $\rho^0$ cells derived from 143B osteosarcoma cells lacking mitochondrial (mt) DNA, and therefore devoid of subunits a and A6L[40].

Here, we show that peripheral stalk subunits are essential to turn the F-ATP synthase into the PTP and that the ANT provides mitochondria with a distinct permeability pathway. Our results resolve a number of outstanding questions about PTP formation by F-ATP synthase and about the role of ANT in the occurrence of the PT, and open new perspectives in understanding this central mystery of mitochondrial biology.

## Results

**The permeability transition in HeLa-$\Delta$g cells.** To test its role in PTP formation, we generated HeLa cells where the g subunit of F-ATP synthase had been deleted by CRISPR/Cas9 technology (Fig. 1a; see Supplementary Fig. 1 for the structure of F-ATP synthase). Absence of subunit g also drastically lowered the level of subunit e, which was virtually undetectable (Fig. 1b), indicating that expression of these two proteins is coordinated. Other components of the lateral stalk were also affected by subunit g ablation, with decreased expression of peripheral stalk subunits b, OSCP and f, while the levels of subunit c were normal (Fig. 1b). Clear native-PAGE analysis revealed that in the absence of subunits g (and e) the complex migrated at lower molecular weight, with the appearance of a species (Fig. 1c, asterisk) which may represent a "vestigial" form of the enzyme[28]. Expression of ANT2 and ANT3, the two major isoforms of the translocator expressed in proliferating cells, was not altered (Fig. 1d). Deletion of subunit g had a strong impact on respiration, which was drastically reduced and became insensitive to oligomycin but could be stimulated by FCCP (Fig. 1e). Note that FCCP-stimulated respiration in wild-type (WT) cells was lower than basal, a toxic effect due to the combination with oligomycin, which indeed was not seen with FCCP alone (Supplementary Fig. 2a). Lower respiration of $\Delta$g cells was matched by a dramatic decrease in the expression of respiratory complexes I, III, and IV (Supplementary Fig. 2b), which has also been observed in HAP1 cells after deletion of peripheral stalk subunits and of the c ring[26–28]. A possible explanation is that respiratory chain complex I and F-ATP synthase share the assembly factors TMEM70 and TMEM242, which could be part of a negative regulatory mechanism connecting the levels of complex I to those of F-ATP synthase[41]. Cell growth was also impaired in HeLa-$\Delta$g cells (Supplementary Fig. 2c), a finding that can be explained by the severe defects of respiration and ATP synthesis. $\Delta$g mitochondria exhibited a significant reduction in the fraction undergoing swelling upon treatment with a $Ca^{2+}$ bolus (Fig. 1f), and consistently showed an increased $Ca^{2+}$ retention capacity (CRC) (Supplementary Fig. 2d).

To further explore the effect of subunit g ablation on the PTP, we studied mitochondria in living cells. HeLa cells were loaded (i) with calcein followed by $Co^{2+}$ to quench the cytosolic calcein signal and allow detection of mitochondrial calcein[42] and (ii) with the potentiometric probe TMRM to detect changes of mitochondrial membrane potential[43] (Fig. 2a). CsA-sensitive loss of mitochondrial calcein fluorescence detects the occurrence of the PT even for very short PTP open times[42] while TMRM release requires longer-lasting PTP openings associated with the release of cytochrome c[44]. We added a cell-permeant hexokinase (HK) 2 peptide that displaces HK2 from the outer mitochondrial membrane[45] and rapidly increases mitochondrial matrix $Ca^{2+}$ by direct transfer from the endoplasmic reticulum, causing PTP opening[46,47]. This is a useful tool to selectively increase $Ca^{2+}$ transfer to mitochondria without perturbing ion gradients across all membranes. Within 2 min of peptide addition, only HeLa WT cells underwent rapid loss of calcein and TMRM fluorescence, suggestive of PTP opening that was confirmed by the full protection exerted by CsA (Fig. 2b). In good agreement with resistance to PTP opening, HeLa-$\Delta$g cells maintained unaltered levels of calcein and TMRM fluorescence throughout the recording period. Importantly, (i) expression of HK2 was maintained and even somewhat increased in $\Delta$g cells (Supplementary Fig. 3a), possibly due to a more glycolytic phenotype; and (ii) the rise of mitochondrial $Ca^{2+}$ elicited by HK2 peptide

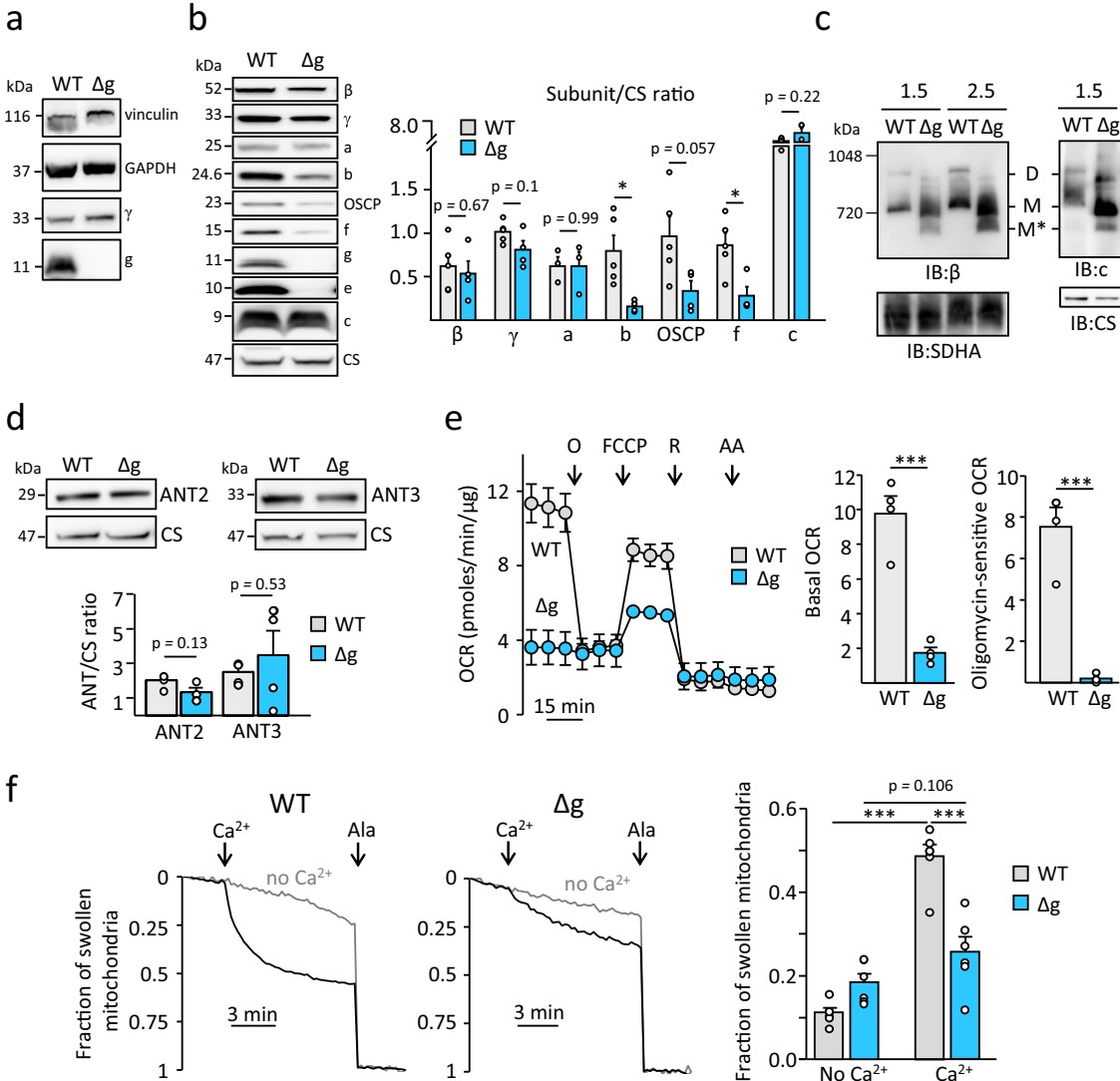

**Fig. 1 Characterization of HeLa-Δg cells. a** Western blot analysis of indicated protein in total cell lysates of HeLa-Δg cells (40 μg/lane). Images are representative of three independent blots. **b** Western blot analysis of isolated mitochondria from wild-type (WT) and Δg cells for the indicated F-ATP synthase subunits. Histogram refers to the quantification of protein levels relative to citrate synthase (CS) and represents mean ± SEM of 3 (for subunit a) or 4 (for all other subunits) independent blots, *p < 0.05, two-sided Student's t-test. Gray bars, WT and cyan bars, HeLa-Δg cells. **c** Clear native-PAGE analysis and subsequent immunoblotting against ATP synthase subunits β, c, and against SDHA of WT and Δg mitochondria in the presence of indicated amount of digitonin (g digitonin/g protein). Images are representative of two independent blots. **d** Western blot on isolated mitochondria for ANT2 and ANT3. Histogram refers to the quantification of protein levels relative to citrate synthase (CS) and represents mean ± SEM of three independent blots. Two-sided Student's t-test. **e** Mitochondrial oxygen consumption rate (OCR) was evaluated in intact cells by Seahorse XF Analyzer before and after the addition of oligomycin (O), FCCP (100 nM), rotenone (R), and antimycin A (AA). Traces are average of four independent experiments for WT (gray trace) and Δg cells (cyan trace). Basal and oligomycin-sensitive OCR is expressed as mean ± SEM of four independent experiments, ***p < 0.001 with two-sided Student's t-test. **f** Swelling assay in isolated mitochondria in the presence (black traces) or absence (red traces) of Ca²⁺. PTP opening was induced with 50 μM Ca²⁺, and alamethicin (ala) was added where indicated. Histograms refer to the fraction of swollen mitochondria after about 9 min of Ca²⁺ addition and are mean ± SEM of six independent experiments, ***p < 0.001, two-sided Student's t-test.

was not significantly different in WT and Δg cells, indicating that resistance to opening was not due to reduced matrix Ca²⁺ load (Supplementary Fig. 3b). The features of the PTP at the single channel level were tested next. High-conductance stable currents with multiple subconductance states were induced by 0.3 mM Ca²⁺ in WT HeLa mitoplasts (i.e., mitochondria devoid of the outer membrane) and these currents were fully blocked by Ba²⁺ (Fig. 2c, left panel and Supplementary Fig. 3c). In striking contrast, under the same conditions, channel activity was never detected in 19 independent experiments with HeLa-Δg mitoplasts, even after increasing [Ca²⁺] to 0.9 mM (Fig. 2c, right panel

and Supplementary Fig. 3d). Channel activity in WT mitoplasts was inhibited by CsA (Supplementary Fig. 3e) but not by BKA (Supplementary Fig. 3f).

The total absence of Ca²⁺-induced channels was surprising, as we would have predicted the appearance of channels mediated by the ANT as observed in HAP1-Δc cells[29]. We therefore tested the effects of ATR and BKA on PT occurrence in cells and PTP opening in patched mitoplasts. As Ca²⁺ is needed for channel formation by ANT[13,14], we treated cells with the ionophore A23187. In these protocols the ionophore was preferred to the HK2 peptide because it allows to calibrate the PTP response to

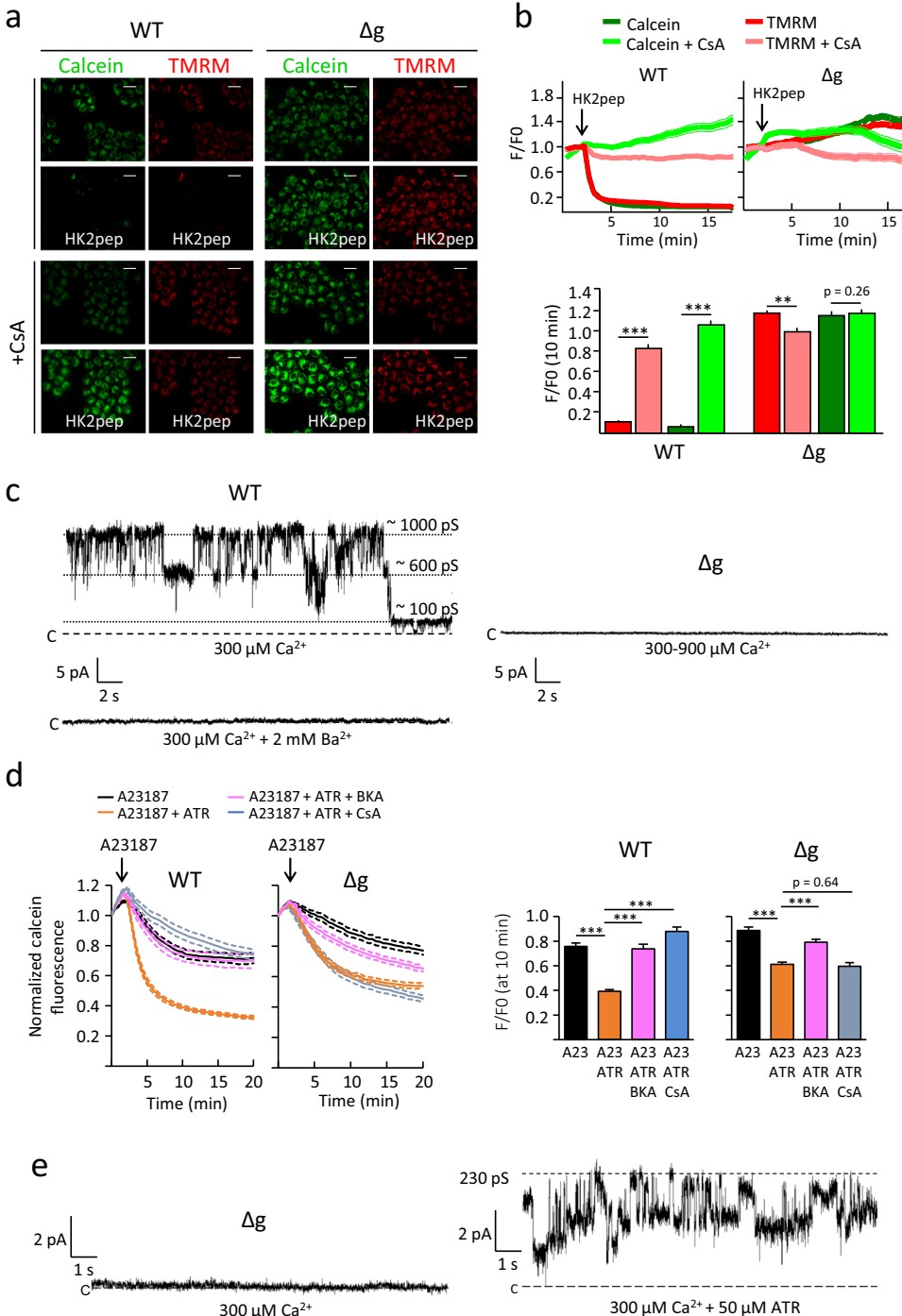

Ca²⁺. We determined the A23187 concentration that does not activate PTP opening per se, and then treated cells with ATR. We confirmed that calcein release is only marginal in Δg cells at 5 μM A23187, i.e., a concentration that readily activates the PTP in WT cells (Supplementary Fig. 3g). Preincubation with ATR significantly accelerated the rate of calcein signal loss after A23187 administration both in WT and Δg cells, a process that was prevented by BKA and CsA in WT cells, and only by BKA in Δg cells (Fig. 2d). Patch-clamp measurements in isolated mitoplasts confirmed the presence of an ATR-induced channel in HeLa-Δg mitoplasts (Fig. 2e).

**The permeability transition in HAP1-Δb and HAP1-ΔOSCP cells**. To study PTP activity in the absence of peripheral stalk

subunits b and OSCP we used HAP1 cells (a kind gift of Prof. Sir John E. Walker), as these have been thoroughly characterized[28] and thus allow a meaningful comparison to be made with the electrophysiological features of the pore, which in these cells have not been tested before. As already seen for subunits g and e, expression of subunits b and OSCP appears to be coordinated. Indeed, Δb mitochondria had a considerable decrease in the expression of subunit OSCP and vice versa (Supplementary Fig. 4a). The expression level of subunits e and g was also dramatically decreased in both mutants (Supplementary Fig. 4a) and these subunits could not be detected in F-ATP synthase complexes in clear-native gels (Supplementary Fig. 4b). We tested the occurrence of the PT in situ in cells loaded with calcein/Co²⁺ and TMRM as before. To our surprise, within 5 min of the addition of

**Fig. 2 HeLa-Δg cells lack the permeability transition but show a latent ANT-related channel. a** Fluorescence microscopy images of WT HeLa and HeLa-Δg cells loaded with 20 nM TMRM and 500 nM calcein-AM/8 mM CoCl$_2$ before and 9 min after the addition of 2.5 μM HK2 peptide (pep) in the absence (top panels) or presence (bottom panels) of 4 μM CsA. Bar, 40 μm. **b** Top: changes in mitochondrial calcein (green traces) and TMRM (red traces) fluorescence intensities in the absence (dark colors) or presence (light colors) of 4 μM CsA. Where indicated, 2.5 μM HK2 peptide (pep) was added. Data are averages of the following ROIs: for WT, 82 (HK2) and 185 (HK2 + CsA) over four independent experiments; for Δg, 101 (HK2) and 133 (HK2 + CsA) over three independent experiments; SEM for each time point is denoted by thin lines. Bottom: Calcein and TMRM fluorescence intensities were analyzed 8 min after peptide addition in the absence (dark colors) or presence (light colors) of 4 μM CsA. Data are average ± SEM of the ROIs indicated above, ***$p < 0.001$, **$p < 0.01$, two-sided Student's $t$-test. **c** Representative current traces showing PTP channel activity obtained by patch-clamping isolated mitoplast from WT and Δg cells in the presence of 300 μM Ca$^{2+}$ ($V_h = +20$ mV). Left panels, currents in WT cells were recorded in 12 experiments out of 17; where indicated, 2 mM Ba$^{2+}$ was added. Right panel, Δg cells with the addition of up to 900 μM Ca$^{2+}$, no current activity was recorded in 19 experiments. **d** Changes in mitochondrial calcein fluorescence in HeLa cells after treatment with A23187 alone (black, 1 μM for WT and 5 μM for Δg), in combination with 50 μM atractylate (ATR, red) or with both ATR and 2 μM bongkrekate (BKA, green) or 2 μM cyclosporin A (CsA, blue). Data are average ± SEM of ROIs as follows: for WT cells, A23187 (88 over four independent experiments), A23187 + ATR (104 over four independent experiments), A23187 + ATR + BKA (72 over three independent experiments) and A23187 + ATR + CsA (78 over three independent experiments); for HeLa-Δg cells, A23187 (158 over eight independent experiments), A23187 + ATR (188 over seven independent experiments), A23187 + ATR + BKA (129 over six independent experiments) and A23187 + ATR + CsA (140 over five independent experiments). For each trace SEM is denoted by thin lines. Histograms refer to calcein fluorescence 8 min after A23187 addition for all conditions, and represent the average ± SEM of the ROIs indicated above, ***$p < 0.001$, two-sided Student's $t$-test. **e** Current traces showing ATR-induced channel activity obtained by patch-clamping isolated mitoplast from HeLa-Δg cells in the presence of 300 μM Ca$^{2+}$ ($V_h = +20$ mV) and ATR (50 μM), which was present both in the pipette and in the bath. The mean values of WT versus Δg + ATR were 827 ± 78 and 620 ± 127 pS ($G_{Max}$), 703 ± 72 and 434 ± 85 pS ($G_{Mean}$), and 48 ± 5 and 47 ± 10 pC ($Q_{4s}$). No statistically significant differences between WT and Δg + ATR were found ($p$ value > 0.17 for all comparisons). Representative currents are from four independent experiments.

HK2 peptide WT, HAP1-Δb, and HAP1-ΔOSCP cells (all of which express HK2, Supplementary Fig. 4c) lost both calcein and TMRM fluorescence (Fig. 3a). Mitochondrial permeabilization was significantly inhibited by CsA in WT but not in HAP1-Δb and HAP1-ΔOSCP cells (Fig. 3a, b and Supplementary Fig. 4d). We then tested the features of the PTP at the single channel level. High-conductance, Ca$^{2+}$-activated stable currents were observed in mitoplasts from all HAP1 cell lines (Fig. 3c), although the frequency was lower in mitoplasts from the deletion mutants. No significant differences were observed in maximal conductance ($G_{max}$), mean conductance ($G_{mean}$), and net charge passing through the channel during its maximal activity ($Q_{4s}$) between WT, Δb, and ΔOSCP mitoplasts (Supplementary Fig. 4e). Consistent with the results in intact cells, HAP1 WT (Fig. 3c, left panel and Supplementary Fig. 5a) but not HAP1-Δb (Fig. 3c, middle panel and Supplementary Fig. 5b) and HAP1-ΔOSCP currents (Fig. 3c, right panel and Supplementary Fig. 5c) were inhibited by CsA, while pore closure rapidly followed the addition of Sr$^{2+}$ (Fig. 3c, middle and right panels), a well-characterized PTP inhibitor[9,10]. Statistical analysis of single channel activity confirmed the total lack of inhibition by CsA in Δb and ΔOSCP mitoplasts (Fig. 3d).

**ANT mediates permeability transition in HAP1 cells lacking OSCP and b subunits.** Given that HAP1-ΔOSCP and HAP1-Δb cells lack lateral stalk subunits (including subunits g and e, which according to the results in HeLa-Δg cells should have caused a loss of PTP activity), the results described in the preceding paragraph were unexpected. It has been reported that HAP1-Δc cells lack a PTP but show a CsA-sensitive channel activated by Ca$^{2+}$ and inhibited by BKA, which might be mediated by ANT[29]. In order to assess whether PTP formation in HAP1-ΔOSCP and HAP1-Δb cells could occur through ANT, we tested the effects of BKA, which does not inhibit channel activity of the purified F-ATP synthase[38]. In intact WT HAP1 cells occurrence of the PT triggered by HK2 detachment was not prevented by preincubation with BKA (Fig. 4a, left panel and Fig. 4b), which instead substantially decreased the mitochondrial calcein and fluorescence loss in HAP1-Δb and HAP1-ΔOSCP cells (Fig. 4a, central and right panel, respectively, and Fig. 4b). Consistent with these findings, in WT mitoplasts currents were insensitive to BKA (Fig. 4c, left panel and Fig. 4d) while BKA blocked the currents in

Δb (Fig. 4c, middle panel and Fig. 4d) and in ΔOSCP mitoplasts (Fig. 4c, right panel and Fig. 4d). Two hundred second recordings are provided in Supplementary Fig. 5. Taken together, these data strongly suggest that both F-ATP synthase and ANT can contribute to the permeability transition in HAP1 cells.

**The permeability transition pore in ρ$^0$ cells.** ρ$^0$ cells lack mtDNA-encoded proteins[40] (Supplementary Fig. 6a) and therefore their ATP synthase lacks the a and A6L subunits. We analyzed calcein and TMRM fluorescence in living cells and the effects of treatment with the HK2 peptide as described above. Although ρ$^0$ mitochondria have no respiratory activity and their ATP synthase does not pump protons, they maintain a membrane potential by hydrolyzing ATP and thus allowing the electrogenic exchange of extramitochondrial ATP for matrix ADP[48,49]. The HK2 peptide caused CsA-sensitive loss of mitochondrial calcein and TMRM fluorescence in both ρ$^+$ and ρ$^0$ cells, consistent with the onset of permeabilization (Fig. 5a, b and Supplementary Fig. 6b) and in keeping with previous results[46]. The time-course analysis revealed that: the process of fluorescence decrease occurred rapidly, being nearly complete within about 2 min of addition of the HK2 peptide; it was very similar for ρ$^+$ and ρ$^0$ cells; and the effect of CsA was somewhat more complete in ρ$^+$ cells, particularly for TMRM (Fig. 5a, b). While these in situ measurements are useful to detect the occurrence of the PT in a population of mitochondria, they do not provide information on whether the absence of subunits a and A6L has affected pore conductance. We therefore tested the features of the pore at the single channel level. The addition of 0.3 mM Ca$^{2+}$ elicited high-conductance stable currents with multiple subconductance states in 11 out of 17 experiments for ρ$^+$ cells and in 10 out of 12 experiments for ρ$^0$ cells (Fig. 5c). In both cell types, currents were completely inhibited by Mg$^{2+}$/ADP (Fig. 5c and Supplementary Fig. 6c, d) and by CsA (Supplementary Fig. 7a) as well as by Ba$^{2+}$ (Supplementary Fig. 7b). On the contrary, both currents were insensitive to BKA (Supplementary Fig. 7c). Statistical analysis of single channel activity revealed no significant differences between ρ$^+$ and ρ$^0$ cells in the maximal or mean conductance, nor in the net charge passing through the channel during its maximal activity (Fig. 5d). We conclude that subunits a and A6L do not contribute to PTP formation.

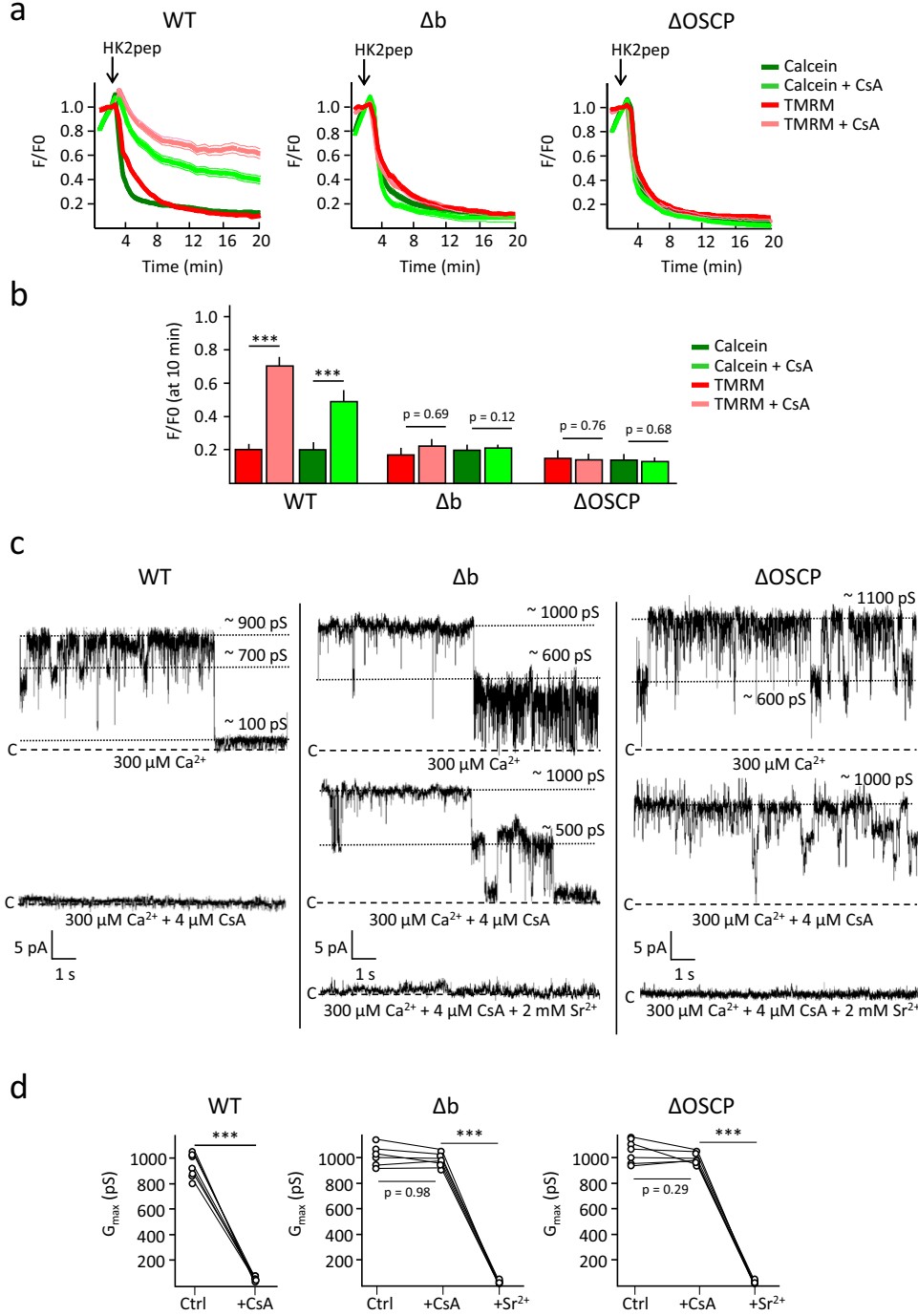

**Fig. 3 The permeability transition in wild-type, Δb, and ΔOSCP HAP1 cells. a** Changes in mitochondrial calcein (green traces) and TMRM (red traces) fluorescence intensities in the absence (dark colors) or presence (light colors) of 4 μM CsA. Where indicated, 15 μM HK2 peptide (pep) was added. Left panel, WT HAP1 cells; middle panel HAP1-Δb cells; right panel, HAP1-ΔOSCP cells. Data are averages of the following ROIs over four independent experiments for each condition and genotype: for WT, 130 (HK2) and 171 (HK2 + CsA); for Δb, 146 (HK2) and 168 (HK2 + CsA); and for ΔOSCP 173 (HK2) and 137 (HK2 + CsA); SEM for each time point is denoted by thin lines. **b** Analysis of the calcein and TMRM fluorescence intensities 8 min after peptide addition in the absence (dark colors) or presence (light colors) of 4 μM CsA. Data are average ± SEM of the ROIs indicated above, ***$p < 0.001$, **$p < 0.01$, two-sided Student's *t*-test. **c** PTP channel activity obtained by patch-clamping isolated mitoplast from WT HAP1 cells (left panel), HAP1-Δb cells (middle panel), and HAP1-ΔOSCP cells (right panel). Where indicated, 4 μM CsA and 2 mM Sr$^{2+}$ were added. Currents in WT cells were recorded in seven experiments out of 11; currents in Δb cells were recorded in seven experiments out of 25; currents in ΔOSCP cells were recorded in eight experiments out of 25. **d** Analysis of the maximal conductance ($G_{max}$) calculated in each independent experiment before and after the addition of 4 μM CsA or 2 mM Sr$^{2+}$. ***$p < 0.001$, **$p < 0.01$, two-sided paired *t*-test.

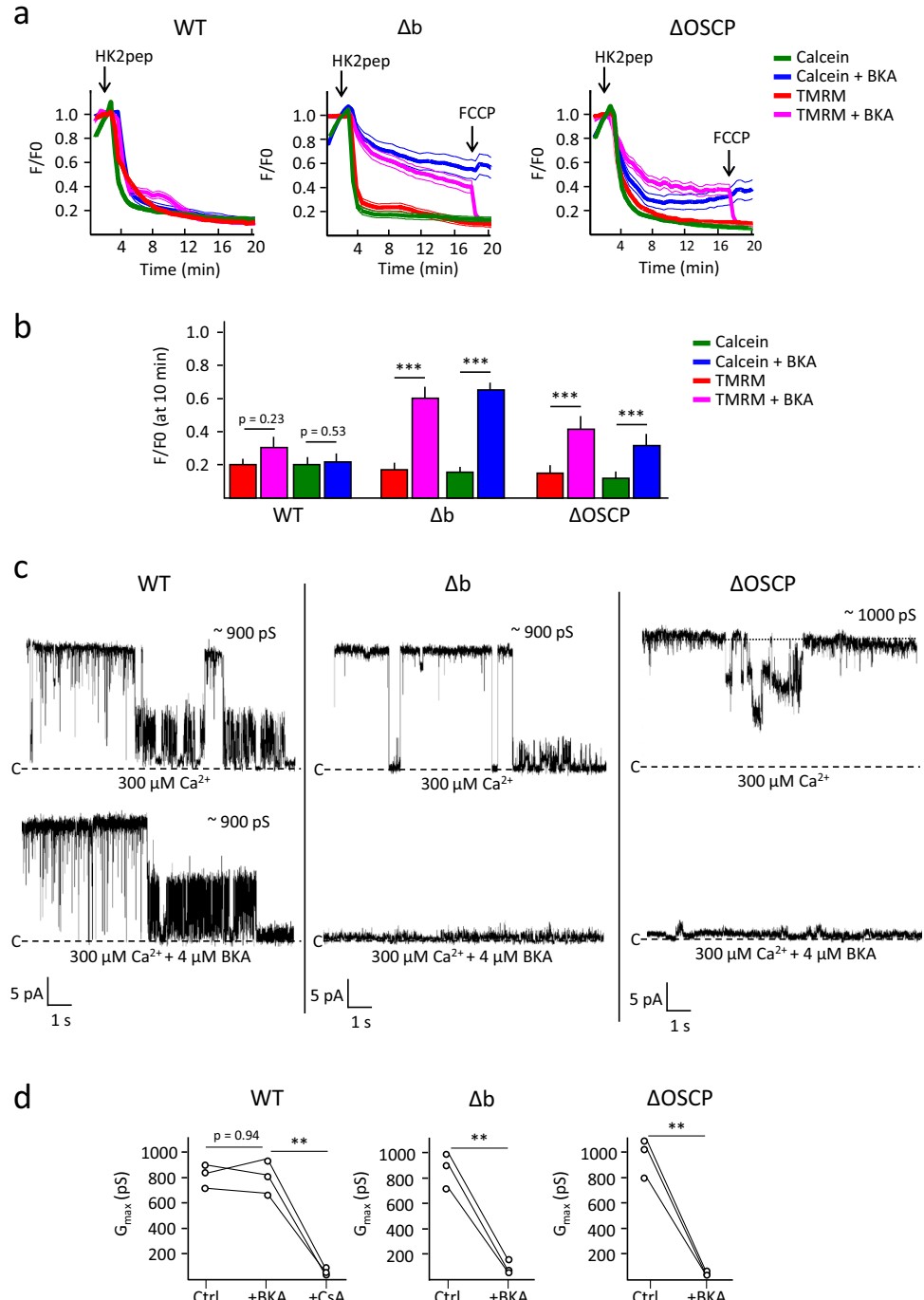

**Fig. 4 Contribution of ANT channel activity to the occurrence of the permeability transition in cells lacking subunits b and OSCP. a** Changes in mitochondrial calcein (green and blue traces) and TMRM (red and purple traces) fluorescence intensities in the absence (green and red) or presence (blue and purple) of 4 μM BKA. Where indicated, 15 μM HK2 peptide (pep) was added. Data are averages of the following ROIs over three independent experiments for each condition and genotype: for WT, 87 (HK2) and 74 (HK2 + BKA); for Δb, 65 (HK2) and 90 (HK2 + BKA); and for ΔOSCP 70 (HK2) and 97 (HK2 + BKA); SEM for each time point is denoted by thin lines. **b** Analysis of the calcein and TMRM fluorescence intensities 8 min after peptide addition in the absence (green and red) or presence (blue and purple) of 4 μM BKA. Data are average ± SEM of the ROIs indicated above, ***$p < 0.001$, **$p < 0.01$, two-sided Student's $t$-test. **c** Channel activity obtained by patch-clamping isolated mitoplast from WT HAP1 cells (left panel), HAP1-Δb cells (middle panel) and HAP1-ΔOSCP cells (right panel). Where indicated, 4 μM BKA was added. Representative currents are from three independent experiments. **d** Analysis of the maximal conductance ($G_{max}$) calculated in each independent experiment before and after the addition of 4 μM BKA or 4 μM CsA. ***$p < 0.001$, **$p < 0.01$, two-sided paired $t$-test.

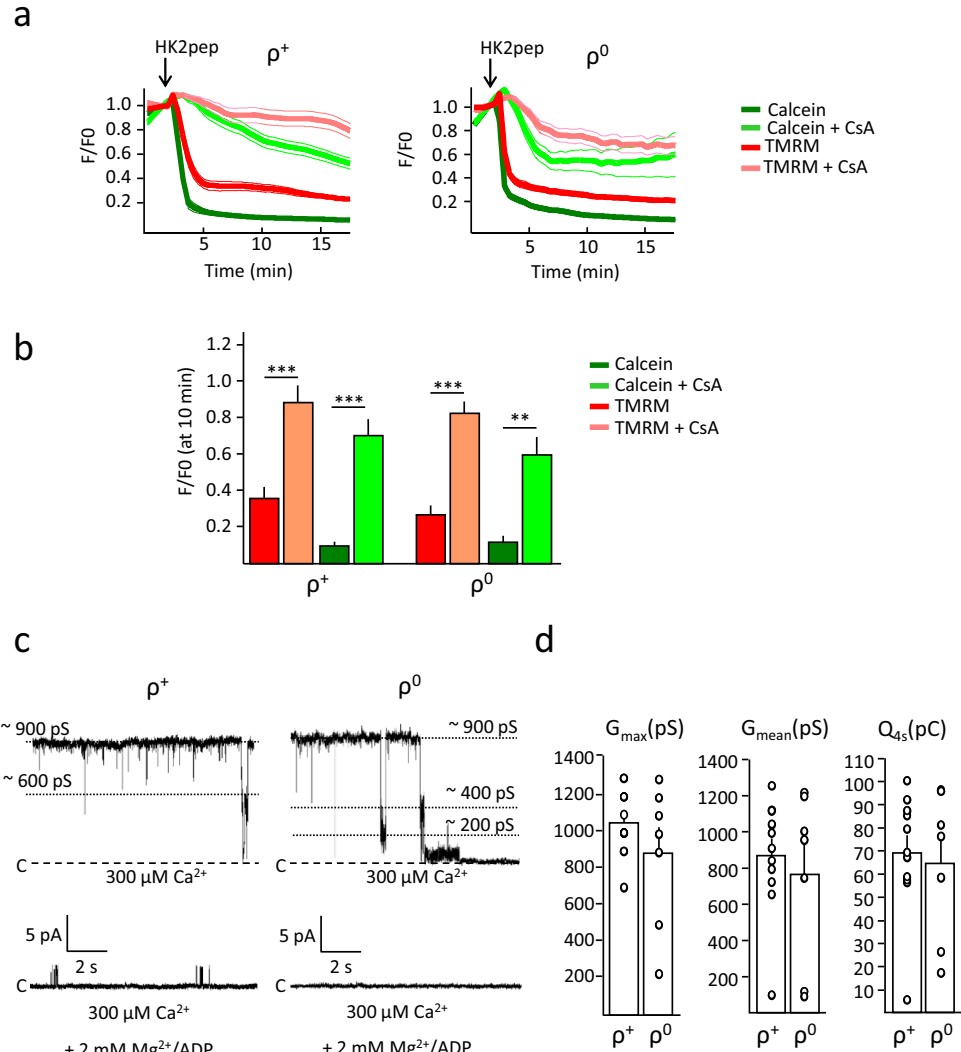

**Fig. 5 The permeability transition pore in $\rho^+$ and $\rho^0$ cells. a** Changes in mitochondrial calcein (green traces) and TMRM (red traces) fluorescence intensities in the absence (dark colors) or presence (light colors) of 4 µM CsA. Where indicated, 2.5 µM HK2 peptide (pep) was added. Data are averages of the following ROIs over three independent experiments for each condition and genotype: for $\rho^+$, 23 (HK2) and 43 (HK2 + CsA); for $\rho^0$, 54 (HK2) and 59 (HK2 + CsA); SEM for each time point is denoted by thin lines. **b** Analysis of the calcein and TMRM fluorescence intensities 8 min after peptide addition in the absence (dark colors) or presence (light colors) of 4 µM CsA. Data are average ± SEM of the ROIs indicated above, $^{***}p < 0.001$, $^{**}p < 0.01$, two-sided Student's $t$-test. **c** Representative current traces showing PTP channel activity obtained by patch-clamping isolated mitoplast from $\rho^+$ and $\rho^0$ cells in the presence of 300 µM Ca$^{2+}$ ($V_h$ = +20 mV). Where indicated, 2 mM Mg$^{2+}$/ADP was present. Currents were recorded in 11 experiments out of 17 in $\rho^+$ cells and in 10 experiments out of 12 in $\rho^0$ cells. **d** Histograms represent the maximal ($G_{max}$) and mean conductance ($G_{mean}$) and the net charge flowing through a stable open channel in an interval of 4 s ($Q_{4s}$) for the currents recorded in $\rho^0$ and $\rho^+$ mitoplasts. Only current traces with events out of the noise range were included in the analysis of $G_{mean}$. Data are average ± SEM.

## Discussion

Whether the PTP originates from a Ca$^{2+}$-dependent conformation of F-ATP synthase is the matter of debate. Evidence in favor is based on reconstitution experiments from mitochondria of various origins[21–24], on knockdown of subunit c[25,29], on generation of point mutations that affect specific channel properties[22,32–37], and on reconstitution of channel activity from highly purified and fully functional F-ATP synthase from bovine and porcine hearts[38,39]. Evidence against is provided by persistence of a CsA-sensitive PT after genetic ablation of subunit c and of constituents of the F-ATP synthase peripheral stalk[26–28]. The findings of the present study provide a solution to this apparent discrepancy and shed new light on the molecular bases of the PT and on the mechanisms of PTP formation.

Our results indicate that in wild-type HeLa, HAP1, and 143B osteosarcoma $\rho^+$ cells, both the Ca$^{2+}$-induced PT in situ and the high-conductance channel recorded by patch-clamp in mitoplasts is inhibited by CsA and unaffected by BKA, the selective inhibitor of the ANT. Given that the channel formed by purified F-ATP synthase is insensitive to BKA[38], we conclude that the PT is mediated by opening of the F-ATP synthase channel (F-PTP). Since both the F-PTP and the ANT channel (A-PTP) are inhibited by CsA, their relative contribution can be inferred from the effects of BKA, which is selective for the A-PTP. Opening of both channels in wild-type cells cannot be excluded, but given the lack of inhibition by BKA we must conclude that in HeLa, HAP1, and $\rho^+$ cells the F-PTP predominates. From these experiments it is clear that cell-specific differences exist, and that the basis for these will need to be addressed in the future, including the extent to

which the use of HK2 peptide compares to ionophores in inducing mitochondrial $Ca^{2+}$ (over)loading.

A second important point is that HeLa-Δg cells, which lack key peripheral stalk subunits and therefore do not have a fully assembled F-ATP synthase, do not undergo a PT nor form high-conductance channels after treatment with $Ca^{2+}$, and therefore lack the F-PTP. Yet, addition of ATR elicits currents indicating that HeLa-Δg cells have a latent, inducible A-PTP. The presence of the latter could provide an explanation to the eventual $Ca^{2+}$ release seen in Δg mitochondria, although the limited respiratory capacity inevitably curbs the ability to accumulate $Ca^{2+}$. The A-PTP was also detected in HAP1-Δb and HAP1-ΔOSCP cells, where prompt BKA-sensitive permeabilization and channel opening followed the addition of HK2 peptide in situ and of $Ca^{2+}$ at the patch-clamp. Thus, while in wild-type HAP1 cells the F-PTP predominates (as indicated by lack of inhibition by BKA), in HAP1-Δb and HAP1-ΔOSCP cells (and at variance from HeLa-Δg cells) activation of the A-PTP does not require ATR. Similar findings were obtained in HAP1-Δc cells, which also displayed currents sensitive to BKA that indicate the presence of the A-PTP[29]. It should be noted that in HeLa-Δg and in HAP1-Δb and HAP1-ΔOSCP cells the A-PTP became insensitive to CsA, a puzzling finding that will be further discussed below.

We also studied $\rho^0$ cells lacking mtDNA[40]. This is a very interesting model because the F-ATP synthase lacks subunits A6L and a, but has an intact peripheral stalk and undergoes full assembly with generation of dimers and oligomers[26,50], although these may have a lower stability toward detergent extraction[50]. Mitochondria in these cells undergo a process of CsA-sensitive permeabilization consistent with PTP opening[26,46] and have CsA-sensitive channels of identical conductance to those of $\rho^+$ cells, as shown here. Thus, F-PTP formation does not require a catalytically active F-ATP synthase, nor $H^+$ transport through the c ring, which cannot occur in the absence of subunit a[51]. It is intriguing that during apoptosis induction $\rho^0$ cells undergo mitochondrial permeabilization in a fashion that is indistinguishable from that of control $\rho^+$ cells[52], which is entirely consistent with preservation of the PT. It will be interesting to test whether PTP-defective cells have an altered response to cell death inducers, although decreased ATP synthesis combined with respiratory inhibition make for a very challenging task.

Our results also bear on the question of how the F-PTP originates from F-ATP synthase. Previous experiments highlighted the importance of an intact peripheral stalk, which is essential for enzyme dimerization[21,38]. In turn, this requirement could explain why we could not detect channel activity in monomers[21,38] and why HeLa-Δg cells lack PTP formation in spite of their normal levels of subunit c. This is in apparent contrast with recent findings reporting channel formation from monomers[39] and leading to the conclusion that the F-PTP forms from the c ring, in keeping with earlier suggestions[22,25]. At variance from the classical PTP, however, channel opening was observed in the absence of added $Ca^{2+}$ and was inhibited by oligomycin[39], which does not block the PTP in native mitochondria[53]. As discussed in more detail elsewhere[19], it is possible that removal of lateral stalk subunits (including e and g) by dodecylmaltoside[39] may have generated an F-PTP that is no longer regulated through subunit e, which directly contacts the lipids within the c ring[54,55]. Thus, both an intact peripheral stalk and the c ring appear to be required for F-PTP formation in situ. The $Ca^{2+}$-induced conformational change would originate at the enzyme crown region[33], and would be transmitted via OSCP and the peripheral stalk to the c ring through the "wedge" or "bundle" region formed by the tight association between the N-termini of e, g, and b subunits[54–56]. This hypothesis, first suggested by Gerle in the "death finger" model for PTP formation and recently revisited[57],

is supported by a deletion study in yeast[32] and by recent structural data on the entire enzyme complex[54].

The sensitivity of both F-PTP and A-PTP to CsA suggests that the PT-promoting regulatory protein CyPD interacts with both F-ATP synthase[20,21] and ANT[14,58]. What remains puzzling is why the channel observed in HAP1-Δb and HAP1-ΔOSCP cells, which is sensitive to BKA and thus mediated by ANT, becomes insensitive to CsA as does the PT induced by ATR in HeLa-Δg cells. The best characterized interaction of CyPD with mitochondrial proteins is with subunit OSCP of F-ATP synthase[21], which has been confirmed in several laboratories[59–62]. Since both HAP1-Δb and HAP1-ΔOSCP cells lack OSCP, this finding suggests that CyPD binding occurs at OSCP and that the presence of F-ATP synthase is required for the ANT to assume the A-PTP conformation, possibly through a physical interaction with F-ATP synthase in the "ATP synthasome", which may also include the Pi carrier[63–65]. Experiments are underway to address this hypothesis. Irrespective of the detailed mechanism through which the F-PTP and A-PTP may communicate, however, the existence of two PTPs provides a convincing explanation for the persistence of $Ca^{2+}$-dependent permeabilization in the absence of an assembled F-ATP synthase[26–28]. We are confident that having solved this apparent discrepancy will further boost research on mitochondrial permeability pathways and on their role in physiology and pathology.

## Methods

**Cell cultures.** The human 143B osteosarcoma ($\rho^+$) and the derived 206 ($\rho^0$) cell lines were kindly provided by Lodovica Vergani (Department of Neurosciences, University of Padova, Padova, Italy). Cells were grown in Dulbecco's modified Eagle medium (DMEM, Gibco) supplemented with 10% fetal bovine serum (FBS, Gibco), 1% penicillin/streptomycin (Pen/Strep, Invitrogen), 50 mg/l uridine, MEM non-essential amino acids (Sigma-Aldrich), and vitamins (Sigma-Aldrich). HeLa cells were cultured in DMEM supplemented with 10% FBS, 1% Pen/Strep, 50 mg/l uridine, MEM non-essential amino acids, and vitamins. HAP1 cells were kindly provided by Sir John E. Walker (Medical Research Council Mitochondrial Biology Unit, University of Cambridge, Cambridge, UK) and cultured in Iscove's Modified Dulbecco's Medium (Gibco) supplemented with 10% FBS and 1% Pen/Strep. Every cell line was cultured in a humidified incubator at 37 °C with 5% $CO_2$.

**Generation of HeLa-Δg cells.** The CRISPR/Cas9 system was used to create HeLa cell lines lacking the expression of ATP5MG gene encoding ATP synthase subunit g. A pair of guide RNAs targeting exon 1 and exon 2 (see Supplementary Table 1) were subcloned into the BbsI site of px330 plasmid (Addgene). HeLa cells were grown in DMEM (Gibco 11965) supplemented with 10% FBS, 100 mg/L uridine, non-essential amino acids (Gibco), and vitamins (Gibco) in a humidified atmosphere of 5% $CO_2$/95% air at 37 °C to 70% confluency in 6-well plates. Cells were then transfected with 6 μl Lipofectamine 2000 with 7 μg px330 gRNA1, 7 μg px330 gRNA2, and 7 μg pAAV Syn-GFP (Addgene). The next day, transfected cells were subjected to FAC sorting based on GFP fluorescence and single cells were placed in individual wells of a 96-well plate. The single colonies were subsequently expanded and the loss of subunit g expression was confirmed by Western blot. For cell growth analysis, $10 \times 10^3$ WT or HeLa-Δg cells were seeded into a 6-well plate and counted after 48, 72, and 96 h.

**Mitochondrial isolation.** Cells grown to 90% confluence were washed twice with cold phosphate-buffered saline (Sigma-Aldrich), detached using a scraper, and centrifuged for 5 min at $600 \times g$. The resulting pellet was resuspended in 2 ml of 250 mM sucrose, 10 mM Tris-HCl, and 100 μM EGTA (pH 7.4). Then, cells were homogenized using a Teflon Potter and the homogenate was centrifuged at $600 \times g$ for 5 min. The resulting supernatant was centrifuged at $7000 \times g$ for 10 min at 4 °C and the pellet containing intact mitochondria was resuspended in 50 μl of the above medium and quantified with the BCA method.

**Mitoplast preparation and patch clamp.** Isolated mitochondria were diluted (1:100) in a solution of 30 mM Tris–HCl, pH 7.4 and let to swell at ice-cold temperature for 10 min to obtain mitoplasts (i.e., mitochondria without the outer membrane). The suspension was then inserted in the patch-clamp chamber and washed with the recording medium. Mitoplasts were well distinguishable from debris, being characterized by a typical cap region (formed by remnants of the outer membrane); mitoplasts suitable for patch clamping, with a diameter of 2–5 μm, were visually selected. Patch-clamp recordings were performed using borosilicate pipettes (5 MΩ) in a solution of 150 mM KCl, 10 mM HEPES, 0.3 mM

$CaCl_2$ (pH 7.4) both in the pipette and in the bath. Giga-ohm seals were established by gentle suction of the membrane section opposite to the cap; the mitoplast membrane, corresponding to mitochondrial inner membrane was maintained intact, leading to a mito-attached configuration. Data were sampled at 10 kHz and filtered at 500 Hz. Single channel currents were monitored at constant holding potential ($V_h$) of +20 mV. Data were acquired at 10 kHz using a L/M EPC-7 amplifier (List-Medical, Darmstadt, Germany), digitized and stored with a Digidata 1322 A and PClamp8.0 acquisition software (all from Molecular Devices). Inducers and inhibitors were added in the bath during the experiment. When indicated, atractyloside (ATR) was presented in the pipette solution and in the bath solution.

**Current analysis**. Data were analyzed using Clampfit software (Molecular Devices) and MATLAB 2007b (MathWorks). Maximal conductance ($G_{max}$) was calculated for every experiment as the maximal transition in channel conductance between two stable states (transition duration <10 ms) detected with a multi-Gaussian fitting of the current amplitude histogram. Mean conductance ($G_{mean}$) was calculated, after offset correction, as the average of the mean conductance measured during channel activity in 30 s before administration of the blocker for each experiment. The $Q_{4s}$ parameter, representing the net charge passing through the fully open channel in a time interval of 4 s, was calculated for each experiment as the integral over 4 s of the current signal at the maximal activity. Statistical comparison of data was assessed with the two-sided Student's $t$-test.

**Live cell imaging**. For epifluorescence microscopy, cells were seeded onto 24 mm diameter round glass coverslips and grown for 1–2 days in the proper culture medium described above. Cells were incubated in DMEM without phenol red (Gibco) plus 0.8 µM cyclosporin H (CsH, Adipogen) to inhibit P-glycoprotein. Mitochondrial membrane potential was monitored with 20 nM tetra-methylrhodamine methyl ester (TMRM, Invitrogen) in combination with 0.5 µM calcein-AM (Invitrogen) and 8 mM $CoCl_2$ to detect PTP openings as described[42]. To test the effect of atractyloside (ATR), HeLa cells were incubated in HBSS (H1387 SIGMA) supplemented with CsH for 30 min with 2 mM $CoCl_2$ and for another 10 min with 0.5 µM calcein-AM. When indicated, cells were incubated since the beginning with 50 µM ATR alone or in combination with 2 µM BKA. After calcein-AM loading, cells were washed with PBS and incubated with HBSS devoid of $CoCl_2$. After 1 min of calcein-AM fluorescence recording, the $Ca^{2+}$ ionophore A23187 was added as indicated in figure legends. Recordings were performed with a DMI6000B inverted microscope (Leica, HCX Plan Apo 40x oil objective, NA 1.25), while keeping cells in the incubation solution. TMRM was excited using an EL6000 lamp (Leica) combined with a 540–580 nm bandpass optical filter and a 595 nm dichroic mirror to reflect the light beam. Emission light passed through the 595 nm dichroic mirror and a 607–683 nm bandpass optical filter. Calcein was excited using the aforementioned lamp combined with a 460–500 nm bandpass optical filter and a 505 nm dichroic mirror. Emission light passed through the 505 nm dichroic mirror and a 512–542 nm bandpass optical filter. Emissions were collected by a DMC4500 CCD camera (Leica). Fluorescence emission was sampled every 30 s using LAS AF software (Leica). After background subtraction, images were analyzed with ImageJ, calculating the fluorescence emissions generated by exciting cells at 480 and 560 nm, respectively, in specific regions of interest (ROIs) comprising the entire mitochondrial network. For GCAMP6f $Ca^{2+}$ measurements, cells were transfected with a cDNA encoding mitochondrial and nuclear GCAMP6f[66]. To perform $Ca^{2+}$ measurements, medium was replaced with DMEM without phenol red supplemented with 0.8 µM CsH (Adipogen) and with 1 mM $CaCl_2$. Fluorescence was recorded with an inverted microscope (Zeiss Axiovert 100, Fluar 40x oil objective, NA 1.30) in the 500–530 nm range (by a bandpass filter, Chroma Technologies). Probes were sequentially excited at 475 and 410 nm, respectively, for 180 and 300 ms, every 5 s. Excitation light produced by a monochromator (polychrome V; TILL Photonics) was filtered with a 505 nm DRLP filter (Chroma Technologies). After background subtraction, images were analyzed with ImageJ, calculating the ratio (R) between emissions generated by exciting cells at 475 and 410 nm, respectively, in specific ROIs comprising the entire mitochondrial network. Standard error of the mean for the signal is denoted by the dashed traces above and below the solid lines.

**$Ca^{2+}$ retention capacity (CRC) and mitochondrial swelling**. The CRC was evaluated with Calcium Green-5N (Molecular Probes) using a Fluoroskan Ascent FL (Thermo Electron) plate reader. Isolated mitochondria were resuspended to a final concentration of 0.4 mg/ml in 130 mM KCl, 10 mM MOPS-Tris, 10 µM EGTA-Tris, pH 7.4, 5 mM glutamate, 2.5 mM malate, 1 mM Pi, and 0.5 µM Calcium Green-5N and then subjected to a train of 2.5 µM $Ca^{2+}$ pulses. Swelling of isolated mitochondria was evaluated by measuring the absorbance at 540 nm using an Infinite M200Pro (Tecan) plate reader. Briefly, 80 µg of mitochondria were resuspended in 130 mM KCl, 10 mM MOPS-Tris, 10 µM EGTA-Tris, pH 7.4 supplemented with 5 mM glutamate, 2.5 mM malate, and 1 mM Pi. PTP opening was triggered by the addition of 50 µM $Ca^{2+}$. At the end of the experiment, 10 µM alamethicin was added to measure maximal mitochondrial swelling. The fraction of swollen mitochondria was calculated as described[53,67]. Statistical comparison of data was assessed with the two-sided Student's $t$-test.

**Oxygen consumption rate**. Oxygen consumption rate in adherent cells was measured with an XF24 Extracellular Flux Analyzer (Seahorse Bioscience). Briefly, HeLa cells were seeded in XF24 microplates at $3 \times 10^4$ cells/well for WT and at $5 \times 10^4$ cells/well for $\Delta$g cells in 200 µl supplemented DMEM and grown at 37 °C in a 5% $CO_2$ humidified incubator for 24 h. Before starting the assay, the growth medium was replaced with Seahorse medium (DMEM-Sigma D5030) supplemented with 143 mM NaCl, 25 mM glucose, 10 mM sodium pyruvate, 2 mM glutamine, and 15 mg/l phenol red. Cells were incubated at 37 °C for 30 min to allow temperature and pH equilibration. After an oxygen consumption rate (OCR) baseline measurement, 1 µg/ml oligomycin, 100 nM FCCP, 1 µM rotenone, and 1 µM antimycin were sequentially added to each well. OCR values were normalized for the protein content and rotenone- and antimycin-insensitive respiration was subtracted. Statistical comparison of data was assessed with the two-sided Student's $t$-test.

**Western blot**. For Western blot analysis, the following antibodies were used in a 1:1000 dilution: anti-β (ab14730, Abcam), anti-γ (PA5-29975, ThermoFisher), anti-b (ab117991, Abcam), anti-OSCP (ab110276, Abcam), anti-f (ab200715, Abcam), anti-g (ab126181, Abcam), anti-e (ab122241, Abcam), anti-c (ab181243, Abcam), anti-a (ab192423, Abcam), anti-citrate synthase (ab96600, Abcam), anti-vinculin (V4505, Sigma), anti-prohibitin (MS-261-P1, NeoMarkers), anti-GAPDH (2118, Cell Signaling), anti-CyPD (ab110324, Abcam), anti-ANT2 (14671, Cell Signaling), anti-ANT3 (PA5-35113, ThermoFischer), anti-OXPHOS (ab110411, Abcam), anti-HKII (sc130358, Santa Cruz), anti-UQCRC1 (sc65238, Santa Cruz), anti-Grim19 (sc271013, Santa Cruz), and anti-SDHA (sc166947, Santa Cruz). Statistical comparison of data was assessed with the two-sided Student's $t$-test.

**Clear native-PAGE**. Clear native-PAGE was performed according to a published protocol[68]. Briefly, isolated mitochondria were resuspended in 50 mM NaCl, 50 mM imidazole/HCl, 2 mM aminocaproic acid, 1 mM EDTA, pH 7.0 at a final concentration of 10 µg/µl, supplemented with the indicated amount of digitonin and subjected to an ultraspin at 100,000$g$ for 25 min at 4 °C. The resulting supernatant was collected and supplemented with 5% glycerol and 0.001% Ponceau S solution. Samples were loaded onto a Native-PAGE 3–12% gel and run in the presence of 50 mM Tricine, 7.5 mM imidazole, pH 7.0 cathode, supplemented with 0.05% deoxycholic acid sodium salt (DOC) and 0.01% $n$-Dodecyl β-D-maltoside (DDM). Samples were then transferred to a PVDF membrane and subjected to western blot analysis for subunit β, c, g, e, and SDHA as indicated in the figure legends.

**Software**. The following software have been used for data acquisition and analysis: pClamp8.0 (Molecular Devices), LAS AS (Leica), Seahorse Wave, Clampfit (Molecular Devices), Origin, Excel, MATLAB 2007b (MathWorks), and ImageJ.

**Reporting summary**. Further information on research design is available in the Nature Research Reporting Summary linked to this article.

## Data availability

Data supporting the findings of this manuscript are available within the article, the Supplementary Information and the Source data files or are available from the corresponding authors. A reporting summary for this article is available as a Supplementary information file. Source data are provided with this paper.

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

## Acknowledgements

This work is in partial fulfilment of the requirements for a Ph.D. to A.C. Supported by AIRC (grants IG23129 to P.B. and IG20286 to I.S.), Fondation Leducq (16CVD04 to P.B. and M.F.), PRIN (2017LHFW42 to P.B. and 20174TB8KW_004 to I.S.). HAP1 cells lacking ATP synthase subunit b or subunit OSCP were provided by Professor Sir John E. Walker.

## Author contributions

A.C., M.C., and P.B. conceived the work. A.C., L.T., M.C., F.C., J.Š., and R.F. carried out experiments. P.B., I.S., and M.F. acquired funds. A.U., M.F., A.R., I.S., M.C., and P.B. supervised the work. A.C., M.C., and P.B. wrote the initial draft. P.B., M.F., A.R., and I.S. reviewed and edited the manuscript.

## Competing interests

The authors declare no competing interests.
