## [Peer Review File. · Nature Communications]

REVIEWER COMMENTS

Reviewer #1 (Remarks to the Author):

This is a very interesting manuscript that answers some questions about a problem that has vexed mitochondrial biology studies for over 50 years. It is submitted by a group that has been at the forefront of this research. They have used three cell lines to study deletion of various subunits of mammalian ATP synthase to determine effects on the mitochondrial permeability transition pore (PTP) using standard cell biology techniques as well as patch clamping of mitoplasts. The data suggest that deletion of different subunits of the stator and the F_0 component of ATP synthase have differential effects on the presence of the PTP and its control by a pore of lower conductance produced by the adenine nucleotide translocase (ANT). These data shed new light on the identity of the PTP, concentrating on the most recent model of it being derived from ATP synthase, but still do not provide conclusive evidence of the actual identity. However, this does not detract from the paper, and it is important to publish this work as it advances the field and leads us closer to a holy grail of mitochondrial biology. Statistical analysis does seem appropriate. There are a few issues to address:

1. Since you have deleted function of ATP synthase, does this affect the health of the cells? This has been characterized in the HAP1 cells by the Walker group and in ρ_0 cells, but what about the HeLa cells?
2. It is interesting the deletion of the d subunit has dramatic effects on respiration presumably due to down regulation of the ETC, as indicated in Figures 1e and S2b. However, can you speculate why the ETC is affected by this deletion? In addition, why is basal respiration higher than uncoupled respiration in WT HeLa cells and why is there no uncoupled respiration in the Δg cells (Figure 1e).?
3. The experiment in Figure 2c does show slight differences in the response between WT and Δg cells. Could this indicate any significant differences in these two cell lines?
4. The paper by Neginskaya (reference 29) has direct relevance to the data presented here, yet is it barely mentioned in the discussion. Please discuss more its relevance to your work.
5. The lower portions of the native immunoblots are not shown. Is there labeling of other bands in these blots?
6. In the left and middle panels of Figure 1f, please label WT and Δg data.
7. In the legend for Figure S2, third line from the bottom, there is a type: "A23" is present before "(gray line)".
8. The legend for Figure S3 seems incorrect, as panel e is of Δb and $\Delta OSCP$ cells and not ρ_0 cells as stated in the legend.

George A. Porter, Jr., MD, PhD

Reviewer #2 (Remarks to the Author):

The paper by Carrer et al. deals with the relative contribution of F-ATP synthase and adenine nucleotide translocator (ANT) to the formation of the mitochondrial permeability transition pore (PTP) using, on one hand, genetic methods to ablate F-ATP subunits (by knockout of nuclear gene-encoded F-ATP subunits or by the ablation of the entire mitochondrial genome, which encode other subunits of F-ATP) and pharmacological modulators of ANT (namely the activator of ANT-dependent channels atractyloside, as well as the inhibitor bongrekate). The authors provide evidence suggesting that specific F-ATP synthase subunits are required for cyclosporine A-sensitive PTP opening and that, in the absence of such subunits, ANT can come into action to generate a permeability pore. The authors may consider improving the paper in the following points:

- Immunoblot comparisons of the abundance of F-ATP synthase subunits should be subjected to quantification and statistical analyses (Fig. 1a,b).

- Fluorescence microphotographs lack size bars (Fig. 2a, Fig. 6A).
- Traces of fluorescence changes over time lack quantification and statistical analyses (Fig. 2b,c,e; Fig. 3a; Fig. 4a; Fig. 5b).
- Patch clamp recording require indications on the number of experiments that have been performed, quantification and statistical analyses (Fig. 2d,f; Fig. 4b).
- One major limitation of the study that some experiments use HeLa (cervical carcinoma) cells while others take advantage of HAP1 (close-to-haploid chronic myeloid cells) or 143B (osteosarcoma) cells, using distinct genetic manipulations for both cell types, which cannot be compared in a direct fashion. This limitation can be overcome by conducting the same types of genetic manipulation in both cell types. At least, this limitation should be clearly mentioned in the Discussion of the paper.
- It is known that F-ATPase and ANT (together with the phosphate carrier) physically and functionally interact to form the "ATP synthasome", perhaps explaining the intricate regulation of the PTP. This idea might be touched upon in the Discussion.
- The authors refrained from studying the cell biological consequences of PTP formation in the cells and hence did not quantify event of cell death, apoptosis or necrosis. The manuscript could be considerably improved by trespassing the purely mitochondrio-centric measurements of PTP. At least, some literature suggesting the implication of both cyclosporin and bongrekate-sensitive PTP events in cell death should be discussed while citing the relevant literature. Similarly, it has been described that rho^o cells undergo mitochondrial permeabilization during apoptosis induction in a fashion that is undistinguishable from rho⁺ control cells (PMID: 8616847).

Reviewer #3 (Remarks to the Author):

This is a very important study that provides critical insights into the role of specific groups on the ATPase inactivation of mPTP. Importantly, the authors provide a systematic evaluation of the mPTP activity, both the levels of intact cells and channel activity measured by direct patch-clamp assay. Authors conclude that parts of the ATPase that are probably not directly involved in pore formation are still required for the mPTP activity. Another important finding of the work is confirmation of the ANT pathway of membrane permeabilization, which, as the authors argue, is distinct from the mPTP pathway. This is high-quality work with well-designed experiments. The data largely support the author's conclusions. I have several recommendations which in my view will help to improve the clarity of the presentation and provide stronger support to the author's conclusions.

Specifically:

1. In Seahorse experiments, FCCP concentration needs to be carefully titrated. 100 nM use in the experiment wasn't sufficient to even bring respiration rates back to the basal levels of the wt. This suggests that higher concentration needs to be used. Similarly, in delta-g, if a higher concentration was used – would this stimulate respiration? This seems to be a critical question since, as it looks now, respiratory capacity in the mutant is extremely low and might not be sufficient to support the calcium uptake needed for mPTP activation.
2. Similar to point 1, since bioenergetics might be quite different between two cell types, it is important to confirm that rates of mito Ca²⁺ uptake are comparable between different cell types for both isolated mito (Fig 1f) and intact cells (Fig. 2a). Ideally, authors could provide CRC values for both cell types. Minor, in fig. 1f change in light scattering does not necessarily represent the "fraction of the swollen mitochondria". As such better term can be used for the Y-axis.
3. The use of the HK2 peptide for mPTP induction needs stronger justification and discussion. This is

an important issue since a lot of discussion is centered around the similarities and differences between current work and prior studies which used different ways to induce mPTP. For example, the group of Prof. Sir Walker did an mPTP assay using ferutinin, which loads mitochondria with calcium directly bypassing MCU. On the other hand, Prof. Jonas's group used ionomycin in concentrations which will induce Ca^{2+} rise in cytoplasm, which is transported into the mito bypassing (presumably) the ER. Is it possible that differences in the type and mode of mPTP could be due to the way how Ca^{2+} was loaded into the mito? Would use of ferutinin have caused mPTP in delta-g? What was the rationale to switch to A23 in the experiment with ANT ligands?

4. The overall organization of the patch-clamp data presentation can be improved. Specifically: 1) On all images figure legends indicate that drugs were added "where indicated" but only "before and after" traces are shown. Wherever possible, why not to present complete traces that will show the kinetics to channel closure or opening in response to the stimuli? This information could provide valuable clues regarding the mechanisms of channel-drug interactions. As presented now currents are not very informative and sometimes confusing. For example, at Fig. 2d channel almost completely closed even at the presence of Ca^{2+} , did it reopen back to the high conductance state before Ba^{2+} was added? Note similar patterns at figures 3b wt, 4b "delta-b"; 5c "Rho0" - without presentation of the full trace they all look like channel closed spontaneously before inhibitor was added; 2) Another very important information that authors could provide is the inclusion of the "0 pA" current levels. As presented now we can only see current at "closed" level which is somewhat ambiguous. What were the current values from "c" to true zero? Were these values different between control and mPTP traces? It will be good to have these values either included into the traces or have values plotted as bar graphs (or both); 3) Please include G_{max} bar graphs to all figures, similar to how it is done at the Fig. 5D; 4) Trace at the SI Fig. 4d looks terribly similar to TIM channel with two distinct flickering levels, is it possible to present another trace more closely resembling PTP - "multi-conductance channel"?

REVIEWER COMMENTS

We would like to thank all Reviewers for the general appreciation of our work and for the useful comments. Our point-by-point answers are reported below in red font. All changes made to the text are highlighted in yellow in the manuscript.

Reviewer #1 (Remarks to the Author):

This is a very interesting manuscript that answers some questions about a problem that has vexed mitochondrial biology studies for over 50 years. It is submitted by a group that has been at the forefront of this research. They have used three cell lines to study deletion of various subunits of mammalian ATP synthase to determine effects on the mitochondrial permeability transition pore (PTP) using standard cell biology techniques as well as patch clamping of mitoplasts. The data suggest that deletion of different subunits of the stator and the Fo component of ATP synthase have differential effects on the presence of the PTP and its control by a pore of lower conductance produced by the adenine nucleotide translocase (ANT). These data shed new light on the identity of the PTP, concentrating on the most recent model of it being derived from ATP synthase, but still do not provide conclusive evidence of the actual identity. However, this does not detract from the paper, and it is important to publish this work as it advances the field and leads us closer to a holy grail of mitochondrial biology. Statistical analysis does seem appropriate.

Thank you George for the comments and the appreciation.

There are a few issues to address:

1. Since you have deleted function of ATP synthase, does this affect the health of the cells? This has been characterized in the HAP1 cells by the Walker group and in p0 cells, but what about the HeLa cells?

We have evaluated cell proliferation (Supplementary Fig. 2c), which indicates that growth rate of Δg cells is slower, in line with the defects documented for Δb and $\Delta OSCP$ HAP1 cells by the Walker group. As mentioned in the text, we noticed that Δg cells appear to be more glycolytic, as also suggested by HKII expression (Supplementary Fig. 3a).

2. It is interesting the deletion of the d subunit has dramatic effects on respiration presumably due to down regulation of the ETC, as indicated in Figures 1e and S2b. However, can you speculate why the ETC is affected by this deletion? In addition, why is basal respiration higher than uncoupled respiration in WT HeLa cells and why is there no uncoupled respiration in the Δg cells (Figure 1e)?

After our manuscript was submitted, Sir John Walker has provided a clue for the decreased respiration in cells that do not properly assemble the ATP synthase, i.e. the demonstration that Complex I and ATP synthase share the assembly factors TMEM70 and TMEM242 [Carroll et al (2021) PNAS 118, e2100558118]. One of his proposed explanations, which we favor, is "A second possible explanation of the association of TMEM70 and TMEM242 with MCIA is that they participate in a negative regulatory mechanism connecting the levels of complex I and ATP synthase. According to this proposal, under physiological conditions when ATP demand is high, the mitochondria would respond by increasing the level of ATP synthase, with TMEM70 and TMEM242 both involved in the assembly of the c8-ring and not associated with MCIA, freeing MCIA to

participate in the assembly of complex I. When the demand for ATP synthase and the levels of nascent subunit c are reduced, TMEM70 and TMEM242 would sequester MCIA and thereby reduce the assembly of complex I.” We now quote the paper and the explanation in the first paragraph of the Results section. Please note that a small stimulation of respiration is detectable after FCCP addition (Fig. 1e, we selected another representative recording where this is seen more clearly), which is consistent with the fact that mitochondria can still accumulate Ca^{2+} , as shown in this manuscript (Supplementary Fig. 2d) and in the Walker paper.

3. The experiment in Figure 2c does show slight differences in the response between WT and Δg cells. Could this indicate any significant differences in these two cell lines?

We think that the small difference in the time required to reach the peak matrix Ca^{2+} concentration may depend on the limited respiratory capacity of Δg cells. Yet, the PTP still does not open at 6 minutes, a point in time when matrix Ca^{2+} is identical in WT and Δg cells. We have added a comment in the second paragraph of the discussion.

4. The paper by Neginskaya (reference 29) has direct relevance to the data presented here, yet is it barely mentioned in the discussion. Please discuss more its relevance to your work.

You seem to have missed that this paper was quoted with some emphasis, twice in the Results: “The total absence of Ca^{2+} -induced channels was surprising, as we would have predicted the appearance of channels mediated by the ANT as observed in HAP1- Δc cells²⁹”, a comment that clearly acknowledges our appreciation of this work; and “It has been reported that HAP1- Δc cells lack a PTP but show a CsA-sensitive channel activated by Ca^{2+} and inhibited by BKA, which might be mediated by ANT²⁹”; and twice in the Discussion: “Evidence in favor is based on reconstitution experiments from mitochondria of various origins²¹⁻²⁴, on knockdown of subunit c^{25,29}” and “Similar findings were obtained in HAP1- Δc cells, which also displayed currents sensitive to BKA that indicate the presence of the A-PTP²⁹”.

5. The lower portions of the native immunoblots are not shown. Is there labeling of other bands in these blots?

The whole immunoblot has been provided with the source data file, and there are no additional bands.

6. In the left and middle panels of Figure 1f, please label WT and Δg data.

Done, thank you.

7. In the legend for Figure S2, third line from the bottom, there is a type: “A23” is present before “(gray line)”.

Corrected, thank you.

8. The legend for Figure S3 seems incorrect, as panel e is of Δb and ΔOSCP cells and not p0 cells as stated in the legend.

Thank you for noticing, this mistake has been corrected.

Reviewer #2 (Remarks to the Author):

The paper by Carrer et al. deals with the relative contribution of F-ATP synthase and adenine nucleotide translocator (ANT) to the formation of the mitochondrial permeability transition pore (PTP) using, on one hand, genetic methods to ablate F-ATP subunits (by knockout of nuclear gene-encoded F-ATP subunits or by the ablation of the entire mitochondrial genome, which encode other subunits of F-ATP) and pharmacological modulators of ANT (namely the activator of ANT-dependent channels atractyloside, as well as the inhibitor bongrekate). The authors provide evidence suggesting that specific F-ATP synthase subunits are required for cyclosporine A-sensitive PTP opening and that, in the absence of such subunits, ANT can come into action to generate a permeability pore. The authors may consider improving the paper in the following points:

Thank you very much for the useful comments.

- Immunoblot comparisons of the abundance of F-ATP synthase subunits should be subjected to quantification and statistical analyses (Fig. 1a,b).

The analysis was presented in the original Supplementary Fig. 2a (which is now in main Fig. 1b), where subunits e and g have been omitted because they are undetectable, as should be clear from the gels. For clarity, we moved this quantification in Fig. 1b next to the WB images.

- Fluorescence microphotographs lack size bars (Fig. 2a, Fig. 6A).

We have added the size bars to all images, thank you for noticing.

- Traces of fluorescence changes over time lack quantification and statistical analyses (Fig. 2b,c,e; Fig. 3a; Fig. 4a; Fig. 5b).

The error bars are on each experimental point, sometimes hard to see because they lie within the symbols, as is now made clear in the Methods Section. We have now included the statistical analysis in new bar graphs (current Figs. 2b,2d,3b,4b,5b). For reasons of space, we moved former panel 2c to Supplementary Fig. 3b).

- Patch clamp recording require indications on the number of experiments that have been performed, quantification and statistical analyses (Fig. 2d,f; Fig. 4b).

N was originally indicated in each figure below the current traces. Since this was missed, we have now indicated the number of replicates in the figure legends. Statistical analysis of Fig. 2c,e (former Fig. 2d,f) was not performed due to the total absence of channel opening events in HeLa- Δg mitoplasts without ATR, hence any kind of statistical test returns p value = 0. The mean values of WT versus Δg + ATR were 827 ± 78 pS and 620 ± 127 pS (G_{Max}), 703 ± 72 pS and 434 ± 85 pS (G_{Mean}), and 48 ± 5 pC and 47 ± 10 pC (Q_{45}) as now indicated in the figure legends. No statistically significant differences between WT and Δg + ATR were found (p value > 0.17 for all comparisons). Statistical analysis of Fig. 4c (former Fig. 4b) was performed and shown in Fig. 4d.

- One major limitation of the study that some experiments use HeLa (cervical carcinoma) cells while others take advantage of HAP1 (close-to-haploid chronic myeloid cells) or 143B

(osteosarcoma) cells, using distinct genetic manipulations for both cell types, which cannot be compared in a direct fashion. This limitation can be overcome by conducting the same types of genetic manipulation in both cell types. At least, this limitation should be clearly mentioned in the Discussion of the paper.

We think that the use of different cell types is a strength in some respects (results do not depend on a specific cell type) and a weakness in others (effects of specific cellular features or the methods used to obtain them). HeLa cells are so widely used that it seemed sensible to carry out our genetic ablation there; while the HAP1 cells were obtained from Sir John Walker with the specific purpose of addressing (and, as it turns out, resolving) what appeared to be a controversy. We have added a sentence about the cell-specific differences in the Discussion.

- It is known that F-ATPase and ANT (together with the phosphate carrier) physically and functionally interact to form the "ATP synthasome", perhaps explaining the intricate regulation of the PTP. This idea might be touched upon in the Discussion.

This idea was indeed mentioned in the last paragraph of the Discussion.

- The authors refrained from studying the cell biological consequences of PTP formation in the cells and hence did not quantify event of cell death, apoptosis or necrosis. The manuscript could be considerably improved by trespassing the purely mitochondrio-centric measurements of PTP. At least, some literature suggesting the implication of both cyclosporin and bongrekate-sensitive PTP events in cell death should be discussed while citing the relevant literature. Similarly, it has been described that ρ^0 cells undergo mitochondrial permeabilization during apoptosis induction in a fashion that is undistinguishable from ρ^+ control cells (PMID: 8616847).

Nice point, we have added a comment and reference to the findings of Marchetti et al in ρ^0 cells in the revision. The problem with cell lines lacking subunits of F-ATP synthase is that lack of proper assembly is matched by severe respiratory impairment (now discussed as also suggested by the 1st Reviewer). These cells have quite a bioenergetic problem, and they are not ideal for studies of cell death, which will be rather addressed with site-specific mutants where neither assembly of the enzyme complex nor respiration are compromised. We added the growth curves (Supplementary Fig. 2c) and a comment to stress this limitation.

Reviewer #3 (Remarks to the Author):

This is a very important study that provides critical insights into the role of specific groups on the ATPase inactivation of mPTP. Importantly, the authors provide a systematic evaluation of the mPTP activity, both the levels of intact cells and channel activity measured by direct patch-clamp assay. Authors conclude that parts of the ATPase that are probably not directly involved in pore formation are still required for the mPTP activity. Another important finding of the work is confirmation of the ANT pathway of membrane permeabilization, which, as the authors argue, is distinct from the mPTP pathway. This is high-quality work with well-designed experiments. The data largely support the author's conclusions. I have several recommendations which in my view will help to improve the clarity of the presentation and provide stronger support to the author's conclusions.

Thank you very much for the appreciation and the useful comments.

Specifically:

1. In Seahorse experiments, FCCP concentration needs to be carefully titrated. 100 nM use in the experiment wasn't sufficient to even bring respiration rates back to the basal levels of the wt. This suggests that higher concentration needs to be used. Similarly, in delta-g, if a higher concentration was used – would this stimulate respiration? This seems to be a critical question since, as it looks now, respiratory capacity in the mutant is extremely low and might not be sufficient to support the calcium uptake needed for mPTP activation.

We do routinely determine the optimal FCCP concentration by titration in each cell line, apologies for not having mentioned this data, which is now provided in Supplementary Fig. 2a. As you can see, 100 nM is a good compromise while 150 nM becomes toxic for the Δg cells. The limited stimulation seen in the experiment is a toxic effect due to the combination with oligomycin, which is prominent in some cell types and observed quite frequently, as also addressed in a recent methodological consensus paper [Connolly et al (2019) *Cell Death Differ* 25, 542-572]. This is now explained in the text.

2. Similar to point 1, since bioenergetics might be quite different between two cell types, it is important to confirm that rates of mito Ca^{2+} uptake are comparable between different cell types for both isolated mito (Fig 1f) and intact cells (Fig. 2a). Ideally, authors could provide CRC values for both cell types. Minor, in fig. 1f change in light scattering does not necessarily represent the “fraction of the swollen mitochondria”. As such better term can be used for the Y-axis.

We agree that Ca^{2+} uptake might be affected by the procedure of mitochondrial isolation itself, especially in case of severe mitochondrial defects. In support to the swelling results, we are now providing the requested CRC experiments (Supplementary Fig. 2d) which, in good agreement with PTP inhibition, show an increased Ca^{2+} threshold for Δg mitochondria. The change in light scattering does represent the fraction of swollen mitochondria, as clearly documented by earlier work from our laboratory [Petronilli et al (1994) *J Biol Chem* 268, 21939-21945, see Figs. 1-2 and the corresponding EM pictures in Fig. 4].

3. The use of the HK2 peptide for mPTP induction needs stronger justification and discussion. This is an important issue since a lot of discussion is centered around the similarities and differences between current work and prior studies which used different ways to induce mPTP. For example, the group of Prof. Sir Walker did an mPTP assay using ferutinin, which loads mitochondria with calcium directly bypassing MCU. On the other hand, Prof. Jonas's group used ionomycin in concentrations which will induce Ca^{2+} rise in cytoplasm, which is transported into the mito bypassing (presumably) the ER. Is it possible that differences in the type and mode of mPTP could be due to the way how Ca^{2+} was loaded into the mito? Would use of ferutinin have caused mPTP in delta-g? What was the rationale to switch to A23 in the experiment with ANT ligands?

We have now explained why we used the HK2 peptide, which has been thoroughly validated in our laboratory [Chiara et al (2008) *PLoS One* 3, e1852; Masgras et al (2012) *Biochimica et Biophysica Acta* 1817, 1860-1866; Ciscato et al (2020) *EMBO Rep* e49117] and is increasingly appreciated as a tool to selectively increase Ca^{2+} transfer in mitochondria without perturbing ion gradients across all membranes (as is the case for ionophores). However, please note that in the experiments with ATR we did use A23187 because, unlike the HK2 peptide, it allows to calibrate the mitochondrial Ca^{2+} load and thus to pinpoint differences in Ca^{2+} sensitivity, as now mentioned in the text. The

PTP-inducing effects of A23187 have been extensively characterized by both TMRM and calcein/cobalt fluorescence changes [Petronilli et al (1999) *Biophys J* **76**, 725-734; Petronilli et al (2001) *J Biol Chem* **276**, 12030-12034; Penzo et al (2004) *J Biol Chem* **279**, 25219-25225]. We would like to stress that, rather than relying on a single method, we have measured PTP opening in situ (membrane potential with TMRM, pore opening with Calcein/Cobalt), mitochondrial swelling in isolated mitochondria and single channel openings; and that, irrespective of the method and the triggering agent, all results demonstrated an amazing coherence. In our opinion this is the best guarantee that the conclusions do not depend on the specifics of the system or of the agent(s) used.

4. The overall organization of the patch-clamp data presentation can be improved. Specifically: 1) On all images figure legends indicate that drugs were added "where indicated" but only "before and after" traces are shown. Wherever possible, why not to present complete traces that will show the kinetics to channel closure or opening in response to the stimuli? This information could provide valuable clues regarding the mechanisms of channel-drug interactions. As presented now currents are not very informative and sometimes confusing. For example, at Fig. 2d channel almost completely closed even at the presence of Ca²⁺, did it reopen back to the high conductance state before Ba²⁺ was added? Note similar patterns at figures 3b wt, 4b "delta-b"; 5c "Rho0" – without presentation of the full trace they all look like channel closed spontaneously before inhibitor was added; 2) Another very important information that authors could provide is the inclusion of the "0 pA" current levels. As presented now we can only see current at "closed" level which is somewhat ambiguous. What were the current values from "c" to true zero? Were these values different between control and mPTP traces? It will be good to have these values either included into the traces or have values plotted as bar graphs (or both); 3) Please include G_{max} bar graphs to all figures, similar to how it is done at the Fig. 5D; 4) Trace at the SI Fig. 4d looks terribly similar to TIM channel with two distinct flickering levels, is it possible to present another trace more closely resembling PTP - "multi-conductance channel"?

1) Compressed current traces as well as amplitude histograms are now shown in Supplementary Figs. 3c, 5a-f, 6c,d; the time-point of the additions are shown on the traces. In our experimental conditions the channels operate in multiple-conductance substates (including the closed one). In the main figures we have therefore shown representative traces where this behavior is evident. Channels in control conditions were never observed to be stably closed (except for Δg). The time required to observe stable channel closure following inhibitor addition varies from experiment to experiment, thus we do not take this aspect into account.

2) The 0 pA level is now indicated in the compressed current traces (Supplementary Figures) as well as in the current amplitude histograms. The closed level (denoted as c) is basically never equal to "0 pA" primarily due to the offset and leak. The differences between "c" and "0 pA" are similar for each genotype (between 2 pA and 12 pA) after complete inhibition, hence they are not related to the intrinsic properties of the channels.

3) We now show the G_{max} graphs (including the sensitivity to CsA, BKA and Sr²⁺) in Figs. 3d and 4d. Statistical analysis of Fig. 2c,e (former Fig. 2d,f) was not performed due to the total absence of channel opening events in HeLa-Δg mitoplasts without ATR, hence any kind of statistical test returns *p* value = 0. The mean values of WT versus Δg + ATR were 827 ± 78 pS and 620 ± 127 pS (G_{Max}), 703 ± 72 pS and 434 ± 85 pS (G_{Mean}), and 48 ± 5 pC and 47 ± 10 pC (Q_{4s}) as now indicated in the figure legends. No statistically significant differences between WT and Δg + ATR were found (*p*

value > 0.17 for all comparisons). Statistical analysis of Fig. 4c (former Fig. 4b) is now shown in Fig. 4d.

4) We thank the Reviewer for pointing out that the trace is similar to that recorded for TIM channel. As suggested, we have selected a different trace in Fig. 5c as well as in Supplementary Fig. 7c (former Supplementary Fig. 4d).

REVIEWERS' COMMENTS

Reviewer #1 (Remarks to the Author):

I am satisfied with the responses to the previous reviews (my own and the other reviewers) and the changes made in this manuscript. I think it is an important piece of work and acceptable for publication.

Reviewer #2 (Remarks to the Author):

None

Reviewer #3 (Remarks to the Author):

Authors did a nice job addressing most of this referee's concerns. Overall, paper is much improved especially in the patch-clamp part and will make an important contribution to the field. However, there still some aspects of work and presentation which is this referee's opinion need to be carefully considered and addressed in order to clarify critical points. Specifically:

1. New data on Fig. 1E that shows Seahorse assay is much improved and clearly demonstrate that KO cells mitochondria are robust enough to have spare respiratory capacity. However, data on Fig. S2a need to be carefully revised. Why in titration experiment deltaG FCCP stimulation was only 25% of the basal level while in Fig. 1E it was 100%? Why absolute levels of OCR, which supposed to be normalized by microG of the material are so different between titration and main experiment? Seahorse fluorescent readout is notoriously difficult to translate into absolute values perhaps titration data will be clearer if normalized as 100% for basal for both cell types or in a similar way. These details do not change the overall conclusions, but they are important especially taking into account the bitter controversies in the field and arguments between different groups.
2. Author's response to the HK2 peptide use addresses my concern only partially. Overall, I didn't question the validity of its usage, my suggestion was to point out in discussion that this study used PTP induction method that is fundamentally different from methods used in previous studies. This should be taken into consideration when data from previous work is compared to the current study. Another critical point that needs to be either addressed experimentally or at least acknowledged in discussion is that HK2 induced Ca load of the mitochondria is limited by the amounts of the ER Ca stores. How can authors be certain that ER load in wt and deltaG is the same? ER Ca load is dependent on the SERCA activity, which is ATP dependent pump and deltaG cells are definitely energetically compromised. It is conceivable that they will have problems feeding SERCA and loading ER with Ca. The easiest way to exclude this scenario would be to just confirm mito Ca load with fluorescent probe or to measure ER Ca stores with Fura-2 in response to SERCA inhibitor.
3. Inclusion of the CRC data provides an important additional information and I believe deserves more attention in the discussion. It is striking that CRC increased very moderately despite complete disappearance of the pore (as judged by patch-clamp and swelling assays that were done at much higher Ca concentrations). Can authors discuss possible mechanisms responsible for this "Ca-induced Ca release"? How could these mechanisms fit into the overall picture regarding the molecular nature of the phenomenon?

Reviewer #3 (Remarks to the Author):

Authors did a nice job addressing most of this referee's concerns. Overall, paper is much improved especially in the patch-clamp part and will make an important contribution to the field.

Thank you for the appreciation of our efforts.

However, there still some aspects of work and presentation which is this referee's opinion need to be carefully considered and addressed in order to clarify critical points. Specifically:

1. New data on Fig. 1E that shows Seahorse assay is much improved and clearly demonstrate that KO cells mitochondria are robust enough to have spare respiratory capacity. However, data on Fig. S2a need to be carefully revised. Why in titration experiment deltaG FCCP stimulation was only 25% of the basal level while in Fig. 1E it was 100%? Why absolute levels of OCR, which supposed to be normalized by microG of the material are so different between titration and main experiment? Seahorse fluorescent readout is notoriously difficult to translate into absolute values perhaps titration data will be clearer if normalized as 100% for basal for both cell types or in a similar way. These details do not change the overall conclusions, but they are important especially taking into account the bitter controversies in the field and arguments between different groups.

Thank you for the insightful observations. The variability in the basal respiratory rate of Δg cells is larger than suggested by the examples that we used in original Fig. 1e and Supplementary Fig. 2a. As suggested, we have revised both Figures by using the average of all experiments. We have also added a panel with data normalized to the initial rate in revised Supplementary Fig. 2a.

2. Author's response to the HK2 peptide use addresses my concern only partially. Overall, I didn't question the validity of its usage, my suggestion was to point out in discussion that this study used PTP induction method that is fundamentally different from methods used in previous studies. This should be taken into consideration when data from previous work is compared to the current study.

We have further discussed this possibility.

Another critical point that needs to be either addressed experimentally or at least acknowledged in discussion is that HK2 induced Ca load of the mitochondria is limited by the amounts of the ER Ca stores. How can authors be certain that ER load in wt and deltaG is the same? ER Ca load is dependent on the SERCA activity, which is ATP dependent pump and deltaG cells are definitely energetically compromised. It is conceivable that they will have problems feeding SERCA and loading ER with Ca. The easiest way to exclude this scenario would be to just confirm mito Ca load with fluorescent probe or to measure ER Ca stores with Fura-2 in response to SERCA inhibitor.

As shown by our measurements the rise of mitochondrial Ca^{2+} is similar (and statistically indistinguishable) in wild-type and Δg cells. It is certainly possible that the ER Ca^{2+} load and/or the number of mitochondria-ER contacts is different, yet upon addition of the HK2 peptide the same amount of Ca^{2+} is transferred to the mitochondrial matrix.

3. Inclusion of the CRC data provides an important additional information and I believe deserves more attention in the discussion. It is striking that CRC increased very moderately despite complete disappearance of the pore (as judged by patch-clamp and swelling assays that were done at much higher Ca concentrations). Can authors discuss possible mechanisms responsible for this "Ca-induced Ca release"? How could these mechanisms fit into the overall picture regarding the molecular nature of the phenomenon?

We have now discussed this issue more thoroughly, thank you for the suggestion.